# Emergence of winner-takes-all connectivity paths in random nanowire networks

Hugh G. Manning [1,2], Fabio Niosi[1,2], Claudia Gomes da Rocha [2,3], Allen T. Bellew[1,2], Colin O'Callaghan [2,3], Subhajit Biswas[4], Patrick F. Flowers[5], Benjamin J. Wiley [5], Justin D. Holmes[4], Mauro S. Ferreira[2,3] & John J. Boland[1,2]

Nanowire networks are promising memristive architectures for neuromorphic applications due to their connectivity and neurosynaptic-like behaviours. Here, we demonstrate a self-similar scaling of the conductance of networks and the junctions that comprise them. We show this behavior is an emergent property of any junction-dominated network. A particular class of junctions naturally leads to the emergence of conductance plateaus and a "winner-takes-all" conducting path that spans the entire network, and which we show corresponds to the lowest-energy connectivity path. The memory stored in the conductance state is distributed across the network but encoded in specific connectivity pathways, similar to that found in biological systems. These results are expected to have important implications for development of neuromorphic devices based on reservoir computing.

[1] School of Chemistry, Trinity College Dublin, Dublin 2, Ireland. [2] Centre for Research on Adaptive Nanostructures and Nanodevices (CRANN) & Advanced Materials and Bioengineering Research (AMBER) Centre, Trinity College Dublin, Dublin 2, Ireland. [3] School of Physics, Trinity College Dublin, Dublin 2, Ireland. [4] Materials Chemistry & Analysis Group, School of Chemistry and the Tyndall National Institute, University College Cork, Cork, Ireland. [5] Department of Chemistry, Duke University, Durham 27708 North Carolina, USA. These authors contributed equally: Hugh G. Manning, Fabio Niosi, Claudia Gomes da Rocha. Correspondence and requests for materials should be addressed to J.J.B. (email: jboland@tcd.ie)

The unique properties of nanoscale materials are well established and they have been responsible for numerous scientific and technological breakthroughs in last number of decades. In comparison to their bulk counterparts, nanomaterials often reveal superior physical properties such as higher strength, lighter weight, increased electrical conduction, and greater chemical reactivity. Currently, these properties are exploited through the integration of individual components (dots, wires, and sheets) into devices or from the benefits derived from the assembly of these components into networks and composites. In each case the presence of surface layers—molecules, surfactants, polymers, and native oxides—essential to stabilize these materials during synthesis and processing, represent barriers to physical integration and electrical connectivity. Thermal, mechanical and chemical processes have been employed to minimize these barriers and develop various applications based on metal nanowire networks (NWN)[1,2], including flexible and transparent conductors[3–10], energy storage[11,12] and generator devices[13–15], sensors and memory devices[16,17]. Nanoscale dielectric layers give rise to material-independent ubiquitous behaviors. For example, electrical stressing of junctions between oxide coated wires show resistive switching; polymer coated wires undergo controlled capacitive breakdown, whereas wires coated with semiconducting layers exhibit memristive-like properties[18–24] in which their electrical resistance depends on the history of the applied current or voltage drop. Identical behaviors are found in planar metal–insulator–metal structures[25–27], despite the different geometry and vastly different contact areas. In each case, the conductance level of the device is associated with a specific memory state. For this reason, metal-insulator-metal memristive structures and the complex none-quilibrium dynamics they manifest are central to the development of next-generation memory devices and brain-inspired technologies. These field-driven behaviors have led to speculation about the physical formation of single "winner-takes-all" (WTA) conducting filaments (CF) at the nanoscale junction, where the memory state is encoded in a single conducting pathway[25,28,29]. This contrasts with the case of artificial neural networks, where WTA networks are algorithms in which neurons in a layer (mostly the output) compete with each other for activation until only one neuron wins and becomes active, namely the one associated to the strongest input signal[30–32]. Here, the WTA has a physical representation in which mobile ions in the metal–insulator–metal junction start to cluster in response to the applied electric field, with the largest cluster growing more rapidly than others[33,34]. Once the first CF is formed, the conduction memory channel is established hindering the growth or interference of other filaments or memory channels.

Here, we demonstrate that it is possible to form WTA connectivity paths in macroscale NWN. The existence of WTA paths is critical to establishing independently addressable memory or conductance states in complex systems. We describe the network properties necessary to establish a WTA path and the possibility of addressing nanoscale components within a macroscopic assembly without the need for direct contacts. To demonstrate this capacity, we first explore the relationship between the electrical behaviors of junctions formed between individual nanowires (NWs), with those of macroscopic assemblies of the same junctions present in random NWNs. We find that for both the increase in conductance scales identically with the current compliance limit used to assess the I–V characteristics of the system. This self-similar scaling holds for all nanomaterial systems studied and simulations reveal it is a property of any network where the junctions dominate transport. Remarkably, we find for junctions with particular scaling properties, the associated macroscopic networks exhibit

conductance plateaus at fractions of the quantum conductance level, $\Gamma_0 = 2e^2/h$. These conductance plateaus are indicative of the formation of a single WTA conducting path across the entire network. We demonstrate that WTA paths have the lowest energy of formation and are stable over a finite energy or input current range. Collectively, these results point to a capacity to self-select the lowest energy connectivity pathway within a complex random network, one that is robust and immune to perturbations. These observations are expected to have important implications for example in the area of neuromorphic (brain-like) devices based on reservoir computing[35–37]. The latter makes use of neural network-based strategies for processing time-varying inputs that is highly effective for identification, prediction and classification tasks; while the connectivity structure of the network or reservoir remains fixed, the nodes (the junctions in the case of NWNs) evolve dynamically in response to input signals and collectively define the internal state of the reservoir. This serves to map lower-dimensional input signals onto outputs of higher dimensions, which are then examined by an external readout function. This work contributes to the search for alternative hardware architectures that are based on the neuromorphic paradigm that will bring forward the next breakthrough in computing-based technology in which the classical von Neumann computer design is replaced by architectures that are brain-inspired.

## Results

**Transport characterization of NW systems.** Figure 1a shows an example of the single junction devices fabricated for this work. Four-probe contacts to single junction device are established using e-beam lithography (EBL) and the contacts either side of the junction were electroformed prior to characterizing the junction itself[38]. Figure 1b shows a magnified scanning electron microscopy (SEM) image of the Ag NW junction from Fig. 1a. Fig. 1c displays the data obtained from I-V sweeps on a single Ag NW junction at increasing current compliance levels ($I_c$). NWNs yielded similar curves but at larger applied voltages (cf. Supplementary Figures 1 and 2). Figure 1d shows an Ag NWN which was spray deposited and contacted using EBL. A shadow mask method for depositing electrodes was also used which allowed for networks with sizes of 50–500 μm to be fabricated. Figure 1e compares the scaling behaviors of single nanoscale junctions with that of a network of junctions. Remarkably the conductance in both systems scale according to the same power-law (PL) $\Gamma = A\, I_c^\alpha$ where $\alpha$ is the scaling exponent and $A$ the PL prefactor. Supplementary Table 1 shows $\alpha$ and $A$ values obtained for a wide variety of systems (Ag core TiO$_2$ shell, Cu, Ni, and TiO$_2$ NWs), which are described in detail in Supplementary Note 1. These results point to a similarity in the manner in which single junctions and networks of junctions become activated, i.e., a similarity between filament growth across a nanoscale junction and the formation of current pathways across the macroscopic network. The PL scaling behavior also holds true for junctions formed using a range of NW systems: Ni–NiO, core–shell Ag–TiO$_2$, and Cu–CuO, in addition to a range of planar electrochemical metallization memory devices in the literature (cf. Supplementary Figure 3). It is also important to note that all NWN samples studied in this work are 50–500 μm in size and experience current levels in the nA to μA range so as to avoid junction welding due to Joule heating.

A second striking observation is the presence of plateaus in the network conductance below the quantum of conductance $\Gamma_0 = 2e^2/h$ [cf. Fig. 1e, f]. Conductance plateaus were not observed in the case of single junctions as it is extremely difficult

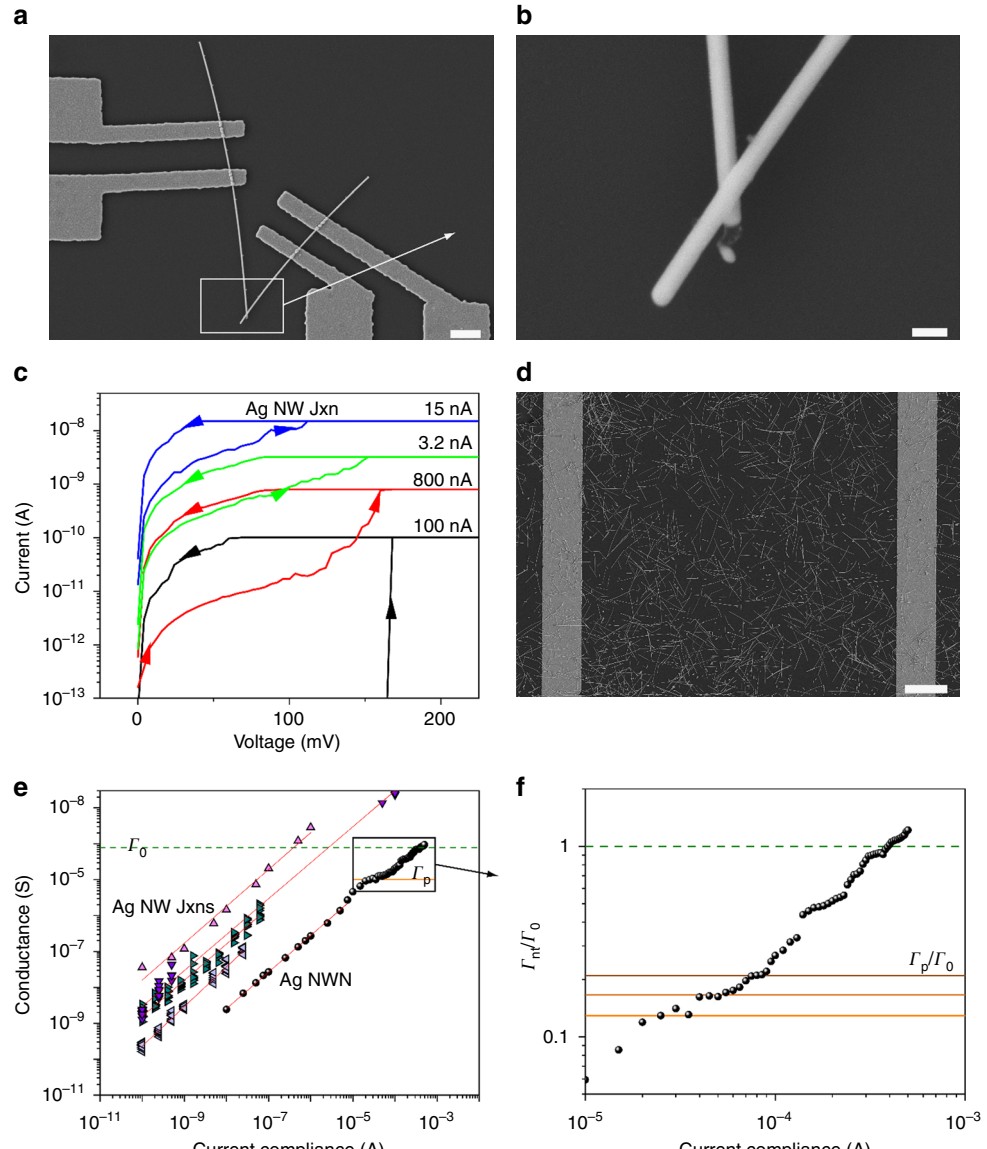

**Fig. 1** Electrical characterization of nanowire systems. **a** Scanning electron microscopy (SEM) image of an Ag nanowire junction (NW Jxn) contacted with four electrodes by electron beam lithography. The white scale bar represents 1 μm. **b** A magnified SEM image of the two overlapping Ag NWs. The scale bar corresponds to 100 nm. **c** I–V curves for different programmed compliance currents for a single Ag NW Jxn. **d** SEM image of an Ag nanowire network (NWN). The white scale bar corresponds to 10 μm. All networks studied here have nearly the same wire density of approximately 0.4 wires/μm$^2$. **e** Conductance plotted against the current compliance (log–log scale) for numerous Ag NW systems. Ag NW Jxns are represented by diamond symbols whereas circles show the measurements for a 500 × 500 μm Ag NWN. The solid lines correspond to power law fits of these datasets. Both systems (Jxn and NWN) display a power-law dependence, however the Ag NWN shows a break from this trend below $\Gamma_0$ (the quantum of conductance) shown as the horizontal dashed green line. Measurement methods are detailed in Supplementary Figures 1 and 2. **f** Zoom-in at the high current compliance range of the Ag NWN conductance ($\Gamma_{nt}$) taken from panel **e** and divided by $\Gamma_0$. This dataset was obtained using an extremely fine current compliance sampling in order to detect the numerous conductance plateaus marked by the solid horizontal lines. The green dashed line depicts the quantum of conductance. The orange horizontal line indicated the first conductance plateau ($\Gamma_p$) found at $\Gamma_p \approx 1 \times 10^{-5}$ S, which gives $\Gamma_p/\Gamma_0 \approx 1/8$. Subsequent plateaus appear at $\Gamma_{nt}/\Gamma_0 \approx 1/6$ and $\Gamma_{nt}/\Gamma_0 \approx 1/5$

to control the conductance and stabilize the formation of single channel CF within a single junction. Not all networks exhibit conductance plateaus, while some networks such as Ag NWNs always exhibit plateaus. We show below that conductance plateaus are observed only for networks with particular junction properties. For networks comprised of this special type of junction, several conductance plateaus are typically found below $\Gamma_0$ as evidenced in Fig. 1f. Further experimentally observed plateaus in Ag and Cu NWNs are shown in Supplementary Figure 4. For each plateau, the network is stable over a range of

input currents, and the connectivity within the network is not significantly affected by the additional power input. While quantum conductance effects has been previously observed in many systems, including a wide variety of CF-based resistive random-access memory devices, percolating nanoparticle films, as well as single- and double-wall carbon nanotube networks[39–43], typically the observation of quantum conductance phenomena requires careful experimental design and precisely controlled electrical measurements. Evidently, networks facilitate detection of subquantum of conductance plateaus even in macroscopic

systems. We show below that the capacity of the network to absorb charge provides a ballast that stabilizes quantum conductance phenomena that occur in localized regions of a macroscopic network.

**Computational model of NWN dynamics**. To explain the origin of the self-similar scaling and the observation of quantized conductance in networks we performed simulations of network transport. To do so we introduce a "part-to-whole" scheme in which we make use of the experimental data gathered for individual junctions (part) to understand the collective behaviour of a network formed by those junctions (whole). The observed PL for the junction conductance ($\Gamma_j$) with current compliance ($I_c$) can be written as:

$$\Gamma_j = A_j I_c^{\alpha_j} \qquad (1)$$

where $A_j$ is a proportionality constant and $\alpha_j$ is a positive exponent that fluctuates around 1 (cf. Supplementary Table 1). By linking this PL behaviour with the memristor charge-carrier drift model, we show that junctions are uniquely described by the set of parameters $\{A_j, \alpha_j\}$, with $A_j$ related to the mobility of the diffusing species that build the filament in the dielectric and $\alpha_j$ captures the nonlinearity in the diffusion barrier (cf. Supplementary Note 2). We assume the junction resistance itself is bounded but can have any value between a high resistance state of $R_{off} \sim 10^4 \, k\Omega$ and a low resistance state (LRS) of $R_{on} = 12.9 \, k\Omega$ (equivalent to $1/\Gamma_0$). These resistance cut-offs were chosen based on our measurements that show that some junctions can in fact reach $\Gamma_0$ at sufficiently high currents. A graphical representation of our Power-Law plus Cut-offs (PL + C) junction-model can be found in the Supplementary Figure 5.

Junctions modelled within PL + C fall into three different types: sub-linear ($\alpha_j < 1$), linear ($\alpha_j = 1$) or supra-linear ($\alpha_j > 1$) each associated with a different strengthening rate ($\nu_j$) at which its conductance increases with the current:

$$\nu_j = \frac{d\Gamma_j}{dI_c} = A_j \alpha_j I_c^{\alpha_j - 1} \qquad (2)$$

The exponent $\alpha_j$ determines the rate with which the conductance of an individual junction changes; whether it speeds-up ($\alpha_j > 1$), stays constant ($\alpha_j = 1$), or slows down ($\alpha_j < 1$) in response to increasing current. The prefactor $A_j$ determines how quickly a junction reaches the LRS. Once the junction reaches its LRS, it becomes a regular resistor with fixed resistance of $R_{on}$ (cf. Supplementary Figure 5).

We now demonstrate that the transport properties of NWNs comprised of junctions modelled within PL + C naturally give rise to emergent behaviours such as self-similarity, conductance plateaus and the selective formation of conducting paths that are immune to input perturbations. To describe such NWNs we exploit our multinodal representation[44,45], which accounts for all resistances, including junction resistances ($R_j$) and inner wire resistances ($R_{in}$). The latter are fixed quantities determined as $R_{in} = \rho l / A_c$, where $\rho$ is the wire resistivity, $l$ its segment length, and $A_c$ its cross-sectional area. An initial amount of current $I = I_0$ is sourced at the electrodes and Kirchhoff's circuit equations written in matrix form, $\hat{M}_R \hat{U} = \hat{I}$, are solved where $\hat{M}_R$ is the resistance matrix (its elements are the inverse of the resistance), and the vectors $\hat{U}$ and $\hat{I}$ are the potential at each circuit node and the current injected/drained out of the device, respectively. The current flowing through each junction is mapped onto a pair of

voltage nodes $(n, m)$ and calculated using

$$I_{n,m} = \frac{|U_n - U_m|}{R_j^{n,m}} \qquad (3)$$

For the first iteration, $R_j^{n,m} = R_{off} \, \forall \, (n, m)$ internode pairs. Once $I_{n,m}$ is determined for all junctions, their new conductance state is obtained using the same functional as in equation (1), i.e.

$$\Gamma_j^{n,m} = A_j \left( I_{n,m} \right)^{\alpha_j} \qquad (4)$$

After updating the conductance of all junctions, the total current sourced on the electrodes is incremented as $I \to I + \Delta I$ and the whole procedure of calculating $\Gamma_j^{n,m}$ takes place recursively until $I$ reaches a predefined maximum value of $I_{max}$. Note that all junctions are subjected to the same cut-off limits set by the $[R_{off}, R_{on}]$ window. The sheet conductance of the network ($\Gamma_{nt}$) is then calculated recursively for each current value $I$ and this outcome is used to compare with the experimental curves of $\Gamma_{exp}$ versus $I_c$ [cf. Figure 1e]. The workflow of the algorithm can be found in the Supplementary Figure 6.

**Simulation results and WTA phenomenon**. Figure 2 shows conductance versus current simulations for an Ag NWN of density 0.49 NWs/μm$^2$. The micrograph image of this network is shown in the Supplementary Figure 7. Each curve represents a distinct set of $\{A_j, \alpha_j\}$ values and in each case four distinct conducting regimes are observed: (i) OFF-threshold (OFF), (ii) transient growth (TG), (iii) PL, and (iv) post-PL (PPL). In the OFF-threshold regime, all junctions are in the OFF-state and the network is not distributing enough current to improve their resistances significantly. At a certain critical current, the conductance of the network increases in a nonlinear fashion as junctions begin to improve their resistances (TG regime). Note that the OFF→TG crossover current depends on the choice of $\{A_j, \alpha_j\}$, and consequently on the strengthening rate $\nu_j$. During the PL stage, we fit $\Gamma_{nt} = A_{nt} I^{\alpha_{nt}}$ onto the numerical curves to obtain the network phase space $\{A_{nt}, \alpha_{nt}\}$, which we then compare with the junction phase space $\{A_j, \alpha_j\}$. The results are presented in Table 1 and from which we find that $\alpha_{nt} = \alpha_j$ in each case. Small discrepancies are observed for larger values of $A_j$ but overall, our simulations demonstrate a one-to-one correspondence between $\alpha_j \leftrightarrow \alpha_{nt}$, consistent with self-similar behaviour found in the experiments.

Figure 2 also points to a rich behaviour in the PPL regime. Whereas single junctions simply transform into ordinary resistors, networks are highly interconnected and continue to evolve in response to the applied current. The actual behaviour depends on the value of $\alpha_j$. For sub-linear junctions as shown in Fig. 2a ($\alpha_j < 1$), the curves vary smoothly with an almost imperceptible change in slope at higher current marking the transition PL→PPL. When $\alpha_j$ is equal to 1 [cf. Figure 2b], the curves show fine structures and finally when $\alpha_j$ is larger than 1, discrete conductance plateaus emerge that are strikingly similar to those found experimentally [cf. Figure 1e]. Figure 2d considers a small dispersion in $\alpha_j$ and an average conductance curve was obtained for a configurational ensemble containing ten sets of $\alpha_j$ distributions used to describe the junctions on the network skeleton (see figure caption of Fig. 2). One can see that apart from the overall step-like conductance behaviour being smoothen out, a robust plateau and other numerous discontinuities are preserved in all curves showing that step-like features in the conductance curves are fingerprints of NWNs composed of

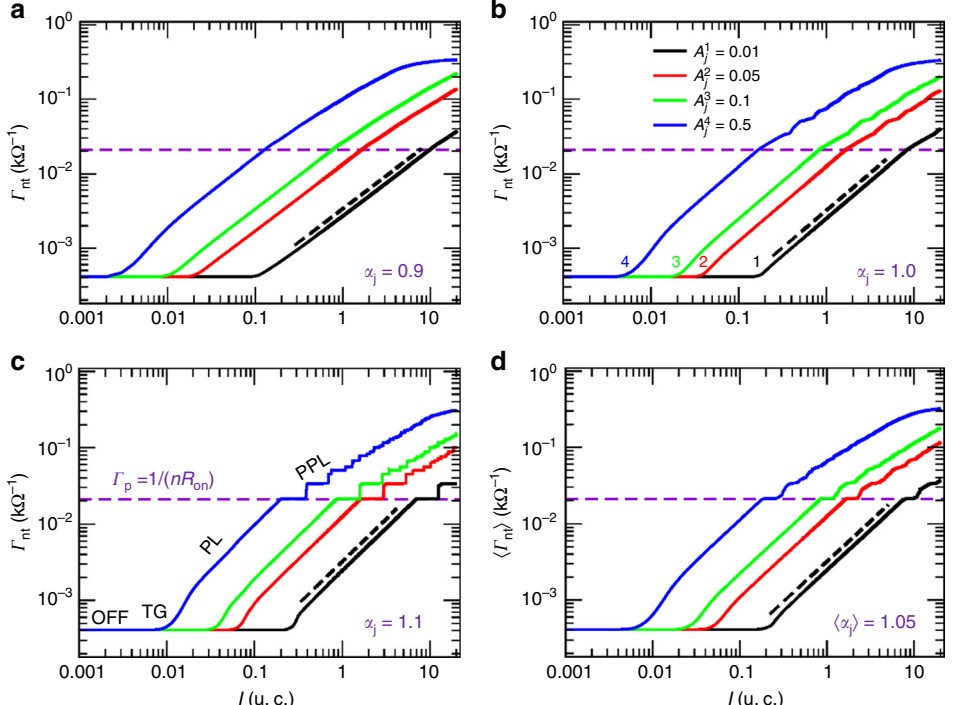

**Fig. 2** Conductance versus current plots taken for an image processed Ag nanowire network. The network wire-density is of 0.49 wires/μm² and its scanning electron microscopy image plus its stick/graph representation can be found in the Supplementary Figure 7. Currents are expressed in units of current (u.c.). The results were taken for distinct values of exponents: **a** $\alpha_j = 0.9$, **b** $\alpha_j = 1$, **c** $\alpha_j = 1.1$, and **d** $\langle\alpha_j\rangle = 1.05$. In the latter, a narrow dispersion was induced in the exponents using a normal distribution with $\langle\alpha_j\rangle = 1.05$, $\sigma = 0.1$ and truncated at [1.0, 1.1]. Each panel contains four curves, one for each $A_j$ value specified on the legend in panel **b**. Numbers "1,2,3,4" on panel **b** label the curve to its corresponding $A_j$. The same labelling scheme and order of the curves hold for all panels. Dashed lines illustrate the power-law fittings that determined $\{A_{nt}, \alpha_{nt}\}$. Results for all fittings are presented on Table 1. Horizontal dashed lines mark the conductance of the first path formed in the network containing $n$ junctions at their optimal state $R_{on}$. This conductance level is given by $\Gamma_p = 1/(nR_{on})$ and for this particular network, $n = 4$. A distinction between the four transport regimes discussed on the main text is depicted on panel **c**: (OFF) OFF-threshold, (TG) transient growth, (PL) power law, and (PPL) post-power-law

**Table 1** $\{A_{nt}, \alpha_{nt}\}$ values for networks obtained by fitting the power law $\Gamma_{nt} = A_{nt}I^{\alpha_{nt}}$ onto the curves of Fig. 2

| $A_j$ | 0.01 | 0.05 | 0.1 | 0.5 |
|---|---|---|---|---|
| $\alpha_j = 0.9$ | {0.0027, 0.892} | {0.0133, 0.896} | {0.0266, 0.9} | {0.1407, 0.925} |
| $\alpha_j = 1.0$ | {0.0025, 1.0} | {0.0125, 1.0} | {0.0251, 1.0} | {0.13071, 1.024} |
| $\alpha_j = 1.1$ | {0.0024, 1.115} | {0.0125, 1.115} | {0.0251, 1.113} | {0.13941, 1.159} |
| $\langle\alpha_j\rangle = 1.05$ | {0.0025, 1.054} | {0.0125, 1.049} | {0.0251, 1.051} | {0.1323, 1.071} |

All junctions in a given network are set to have the same prefactor and exponent except for the heterogeneous case in which a narrow dispersion was induced in the exponents using a truncated normal distribution with mean value of $\alpha_j$. Note the strong correlation between $\alpha_{nt}$ and $\alpha_j$.

supra-linear junctions. We note that Ag junctions considered here are mostly supra-linear with $\alpha_j$ taking values between 1.0 and 1.1.

To explore the precise origin of plateaus in networks with supra-linear junctions, we calculate the amount of current flowing through each wire segment ($I_s$) and plot these data in contour maps in Fig. 3. Movies revealing the complete evolution of the network in response to the current source, junction optimization of the top-three paths of least-resistance, and current contour maps are provided in Supplementary Movies 1 and 2 with a detailed description of these animations found in Supplementary Note 3. The network segment-skeleton that serves as a template for these current maps is shown in the Supplementary Figure 7. Current-contour maps were obtained at distinct source-current values to capture the evolution of the conductance state of the network in the TG, PL, and PPL regimes. These states are

identified in the panel (a) curve by different shaped symbols (square, star, triangle, and circle). The corresponding current maps are displayed in the coloured panels in Fig. 3 and provide a graphical representation of the distinct conducting pathways that emerge naturally as the source current is increased. In the TG regime, the limited input-current spreads across the network to probe the most efficient way to transmit the sourced current. As the input-current increases, the network could, in principle, keep availing of the hundreds of path combinations evidenced in the TG current map to transmit the sourced signal but instead, it selects the "easiest-conducting-path", the WTA path. Once the WTA path is fully optimized (with all its junctions at the LRS when $I = 2.25$ u.c.), the network becomes temporarily Ohmic, reflected by the appearance of the first conductance plateau in the $\Gamma_{nt}$ versus $I$ curve. Since the inner resistance of the metal wires is negligible, the conductance of this first plateau ($\Gamma_p$) is found at

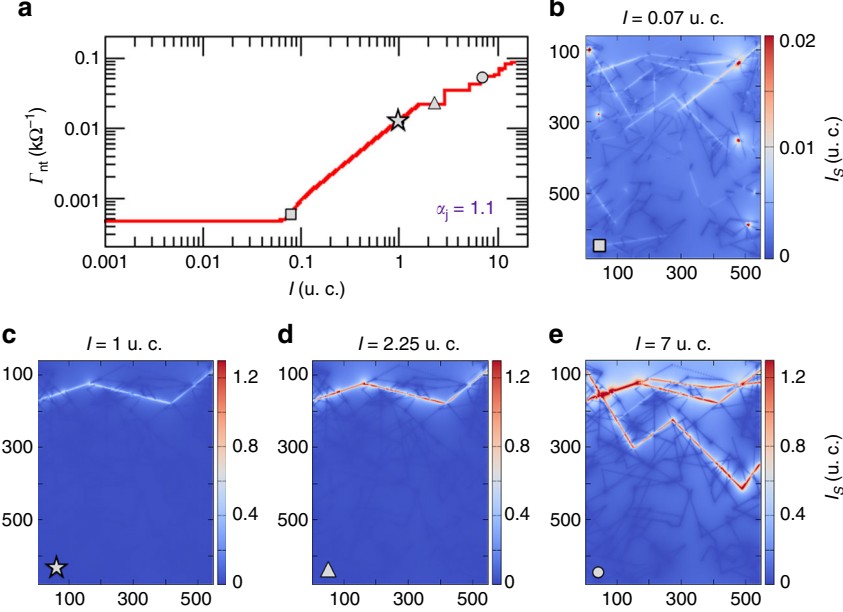

**Fig. 3** Simulated conductance evolution of an Ag nanowire network (NWN). **a** The same conductance versus current curve shown in Fig. 2 for the Ag NWN template depicted in the Supplementary Figure 7. The junction characteristics are set at $A_j = 0.05$ and $\alpha_j = 1.1$. The symbols mark points in the curves in which current colour maps were taken. **b–e** Current colour maps calculated over each wire segment ($I_s$) of the Ag NWN. Snapshots were taken for four sourced current values specified on the top of each current map and distinguished by the symbols: square (transient growth), star (power-law), triangle and circle (both set in the post-power-law regime). In particular, the post-power-law state at $I = 2.25$ u.c. is located at the first conductance plateau, $\Gamma_p = \Gamma_0/4$. Animations revealing the complete evolution of the network in response to the current source, junction optimization of the top-three paths of least-resistance, and current-segment maps are provided in the supplemental material (cf. Supplementary Note 3 and Supplementary Figure 9 for animation description)

approximately $\Gamma_p = 1/(nR_{on}) = \Gamma_0/n$, where $n$ is the number of junctions along the path. This means that each junction along the path behaves as a single conduction channel with an effective resistance of $R_{on} = 1/\Gamma_0$. As more current is pushed through the network, other paths are opened in a discrete fashion leading to discontinuous jumps and additional conductance plateaus. Three well-defined paths (two partially superimposed) can be seen in the contour plot of Fig. 3 with sourced current of $I = 7.0$ u.c.. A similar current map for a different network system is provided in the Supplementary Figure 8 to confirm the generality of this behaviour. Simulation movies for this second network example can be seen in Supplementary Movies 3 and 4 with a detailed description of these animations provided in Supplementary Note 3 and Supplementary Figure 9. These findings are consistent with the experimentally measured conductance-plateaus being located at fractions of $\Gamma_0$ [cf. Figure 1f]. Note that these conductance-plateaus can be observed only when extremely fine current steps are used in the measurements.

The actual WTA path chosen by the NWN is controlled by several factors. The random nature of the network skeleton serves as a first selection mechanism for the propagation of current as it eliminates spatial redundancies. The second selection mechanism relates to the memristive character of the junctions, in particular the value of $\alpha_j$ and consequently the strengthening rate $\nu_j$. Supplementary Figure 10 depicts the current maps for all four conducting regimes addressed in Fig. 3 but it also includes the results for the same network interconnected by linear and sublinear junctions. When $\alpha_j \leq 1$, the network starts its dynamics by selecting two superposing paths that initially have nearly equal least-resistance levels. These two paths are strengthened throughout the PL regime until other paths start to emerge gradually in the PPL stage. The situation changes dramatically when $\alpha_j$ is larger than 1 with the network selecting a single conducting path

in the PL regime. This is a direct consequence of some junctions being optimized faster than others. Under these conditions, the strengthening rate of the junctions described by equation (2) increases with the current. This means that junctions favoured by the network topology will improve at a faster pace whereas those that are hindered by the wire assembly will be delayed in the race for optimal conductance state. This mechanism naturally leads to a WTA outcome similar to that employed in supervised competitive learning in recurrent neural networks[46,47].

The introduction of dispersion in the exponents further drives the network in choosing a single path in comparison to the idealized case in which all junctions have the same characteristics (cf. Supplementary Figure 11). Even paths that nominally have the same initial conductance rapidly differentiate when dispersion is included. Mixing junctions of distinct strengthening rates is a third selection mechanism for the network with some junctions evolving faster than others. The fact that these junctions with slightly different properties are embedded in a highly disordered NW template makes each one of them a unique building block of the network and so redundancies in the propagation of current and path formation are readily eliminated. Deviations from the PL behaviour can occur however when the inner resistance of the wires starts to play an important role. To demonstrate this effect, we artificially increased the resistivity of the wires while keeping the junction resistances fluctuating in the same $[R_{off}, R_{on}]$ range (cf. Supplementary Note 4). These results are shown in the Supplementary Figure 12 and they indicate that a clean PL scaling behaviour and a conducting WTA state can only be found for systems in which $R_{in} \ll R_j$. Other sample characteristics such as NW geometrical aspects, network density, width of the devices, contact resistance (cf. Supplementary Figure 13), and electrode separation are also found to affect the details of PL dynamics and the formation of WTA paths in random NWNs because they all

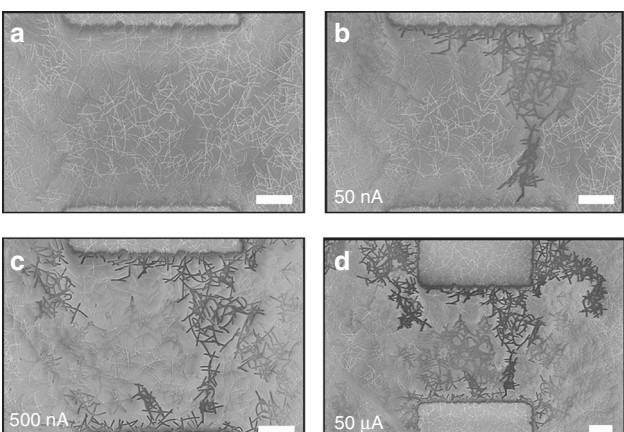

**Fig. 4** Direct visualization of winner-takes-all (WTA) conduction in random nanowire networks (NWNs). **a** Scanning electron microscopy image of an unperturbed Ag NWN of dimensions $100 \times 100\,\mu m$. **b–d** Passive voltage contrast images of the same network taken during I–V sweeps with limiting current compliances of **b** 50 nA, **c** 500 nA, and **d** 50 μA. Note it is not possible to directly compare the contrast observed in different networks or even that observed in the same network imaged under different conditions. Current levels are written on the respective panels. White scale bars correspond to 2 μm. The respective conductance values measured from the I–V curves are **b** $\Gamma_{nt} = 1.1 \times 10^{-7}\,S$, **c** $\Gamma_{nt} = 1.0 \times 10^{-6}\,S$, and **d** $\Gamma_{nt} = 9.6 \times 10^{-6}\,S$

have an impact on the network connectivity[48–50]. That said, any network for which $R_{in} \ll R_j$ and $\alpha_j > 1$ tends to exhibit a WTA path.

We now demonstrate that the self-selective behaviour of the network is energy-efficient. The power consumed by the network is given by $P_{nt} = I^2/\Gamma_{nt}$ and in the PL regime since $\Gamma_{nt} = A_{nt}I^{\alpha_{nt}}$, we can write

$$P_{nt} = \frac{I^{2-\alpha_{nt}}}{A_{nt}} \qquad (7)$$

so that the power varies nonquadratically with the current. Since the network selects a path with the largest exponent $\alpha_{nt}$, equation (7) insures that the establishment of this path dissipates the lowest possible power. As more current is pushed through, the network enters into the PPL regime and a conductance plateau is formed. Since $\alpha_{nt} = 0$ in the plateau region, $P_{nt} = I^2/A_{nt}$ and hence the network becomes Ohmic. In the limit of sufficiently high currents, and after the emergence of successive conductance plateaus, one expects that almost 100% of the junctions will be fully optimized and the network reaches saturation with $\Gamma_{nt} \rightarrow$ constant and $P_{nt} \propto I^2$.

To confirm the simulation outcomes and to directly visualise WTA paths, passive voltage contrast (PVC) SEM images were acquired in Ag NWNs by grounding the electrodes at selected points during a conductance versus current-compliance measurement. Wires that are connected to a grounded electrode through electrical activation appear darker, whereas disconnected or less well-connected wires appear brighter, so that a PVC image can provide a qualitative comparison of the electrical connectivity within a given network. Figure 4a shows the unperturbed network with no contrasting wires in the image. The current compliance was then gradually ramped up and at $I_c = 50$ nA in Fig. 4b we observe the emergence of a WTA path, that is, about to bridge the bottom electrode. On increasing the current, the path is established and then reinforced; at $I_c = 500$ nA shown in Fig. 4c those wires taking part in the conduction dominate the contrast.

As the compliance current is increased further, the path is strengthened as evidence by the increased relative contrast, until finally secondary paths begin to emerge in Fig. 4d as two extra percolating paths emerge from the top electrode. PVC measurements performed on larger NWN samples are shown in the Supplementary Figure 14. They help visualise the behaviour in the TG and PPL regimes prior and after the WTA has been chosen. In the TG regime (recorded following a current of few pA) large areas of the network show contrast consistent with the simulations in Fig. 3 during which the entire network is uniformly probed prior to selecting the WTA path. Supplementary Figure 14(b) shows the presence of multiple conducting paths in the PPL regime.

## Discussion

In summary, we have described the junction and network properties necessary to establish WTA paths in random NWNs. PL behaviour is expected for any network in which the conductance is controlled by the junction properties so that self-similar scaling of junctions and networks is a natural property of junction dominated networks. However, networks comprised of junctions that strengthen under current flow ($\alpha_j > 1$) lead to the development of a WTA path that exhibits characteristic conductance plateaus which are stable over a current compliance range and represents the lowest possible energy connectivity pathway in the network. More generally, this work shows that the memory state stored in the conductance of the entire network is actually encoded in a specific connectivity pathway within, similar to that found in biological systems. It remains to develop approaches towards the rational design of networks that favor WTA formation. Specifically, further work is needed to understand how to engineer the properties of junctions to better satisfy the supra-linear ($\alpha_j > 1$) requirement. Collectively, these findings should help in the development of hardware neural network systems with brain-inspired architectures for cognitive signal-processing, decision-making systems and ultimately neuromorphic computing applications.

## Methods

**Measurements and materials**. Further information on the NWs used in this study, including transmission electron microscopy (TEM)/SEM images, details of I–V and I–t experiments (t being the time) can be found in the Supplementary Figures 1, 2, and 15. Comparisons between scaling of conductive bridging random-access memory literature data and experimental data from various NW systems can also be found in the Supplementary Figure 3. Experiments were carried out on P-type Silicon wafers (University Wafer) with a 300 nm thermally grown SiO₂; NW solutions were dropcast onto substrates prepatterned by ultraviolet lithography. Single, crossed wires, and isolated NWNs were fabricated using previously reported techniques[44]. Larger NWNs were spray-deposited and contacted using a shadow mask.

**Electron microscopy and electrical measurements**. SEM images and EBL were performed on a Zeiss Supra FEG-SEM. TEM images were acquired using a FEI TITAN TEM. Electrical measurements were carried out at ambient conditions using two setups. A Keithley 4200 Semiconductor characterization station was primarily used for electrical measurements. Experiments were also undertaken on a Keithley 3450 and 6450 femtoamp paired with custom LabView interface.

**PVC experiments**. A Zeiss Ultra FEG-SEM paired with a Keithley 2400 and Kleindiek Nanotechnik probing system was used to perform PVC imaging[51]. The secondary electrons, produced by a low energy electron beam (2–4 kV) were used to visualize parts of the network which were electrically activated and connected to an earth reference. Small apertures were also used to improve the quality of the image reducing the background contrast.

**Data availability**. The data that support the findings of this study are available from the corresponding author upon reasonable request.

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

## Acknowledgements

The authors wish to acknowledge funding from the European Research Council under Advanced Grant 321160. This publication has emanated from research supported in part by the research grants from Science Foundation Ireland (SFI) AMBER Centre under Grant number SFI/12/RC/2278 and SFI ivP Grant (12/IA/1482). The facilities and staff at the Advanced Microscope Laboratory at Trinity College Dublin are greatly acknowledged for their support, as is the Research IT Unit (formerly TCHPC) at Trinity College Dublin for computational resources. The authors which to thank Anurag Gupta for program-ming the LabView Software used for data acquisition, and developing custom test modules for the Keithley SCS-4200.

## Author contributions

H.G.M. cowrote the paper, and along with A.T.B. performed experiments on individual NWs, junctions. F.N. performed measurements on NWN samples and passive voltage experiments. C.G.R. cowrote the paper, and along with C.O.'C. developed the compu-tational model and ran the simulations. S.B. and J.D.H. fabricated Ag–TiO₂ NWs. P.F.F. and B.J.W. fabricated Cu NWs. M.S.F. developed the computational model and led the

computational efforts; and J.J.B. led overall effort and cowrote the paper. All authors discussed and commented on the manuscript and on the results.

## Additional information

**Competing interests:** The authors declare no competing interests.

