## [Peer Review File · Nature Communications]

Reviewers' comments:

Reviewer #2 (Remarks to the Author):

This paper ('Emergence of Winner-takes-all Connectivity Paths in Random Nanowire Networks' by Manning et. al.) presents a very interesting study on the junction and network properties necessary to establish a single WTA conducting path in random nanowire network. This conducting path represents the lowest possible energy paths and enable the establishment of independently addressable memory or conductance states within complex network systems.

Considering the recent tremendous interest in metal nanowire based network, this study can provide a very valuable approach to establish a design principal for the optimum metal nanowire network. This paper should be recommended for publication after reflecting the following suggestion in the revision,

1) As a researcher in metal nanowire network, I have never thought about this kind of WTA conducting path formation. The authors very nicely provided extensive discussion and evidences to support their claims.

2) However, the authors said "These observations are expected to have important implications for the operation of neuromorphic devices based on reservoir computing" without any brief explanation on the 'reservoir computing'. Short explanation on 'reservoir computing' will greatly help the readers better understand the main idea in this paper.

3) Even very tiny amount of current can induce a dramatic temperature rise in a metal nanowire (Small, 10(24), 5015-5022 (2014).). The junction resistance between metal nanowires plays a very critical role. Due to very small size, tiny current flow during measurement step can induce localized resistive heating at the nanowire junction, which leads to the welding effect at junction. Once there is any welding phenomena happens the electrical properties of the metal nanowire junction will change irreversibly. Especially, for the WTA case, there will be current crowding at the junctions along the WTA path and resistive heating effect at the junction and along the nanowires as well (just like the pictures shown in Figure 3). Any discussion on this effect during measurement?

4) Does this WTA conducting path in random nanowires happen to the nanowires network with good welding (thermal annealing)? or only to the NWN without welding?

5) Some studies presented that NW length is very important for the electrical properties of metal nanowire network (Nanoscale, 4, 6408 (2012); Crystal Growth & Design, 12, 5598 (2012)). This WTA phenomena could be dependent on the NW geometry such as NW diameter and NW length and deposited NW density. Any discussion on this?

6) Metal nanowire percolation networks have shown various applications. The detailed example of metal nanowire will be very helpful for readers. The application areas on metal nanowire percolation network [(a) touch panel: Adv. Funct. Mater., 23, 4171 (2013); RSC Advances, 6, 57434 (2016); Nanotechnology, 27, 295201 (2016). (b) stretchable electrode: Adv. Mater., 26, 5808 (2014); Adv. Mater., 24, 3326 (2012); Adv. Mater., 27, 6397 (2015); , (c) energy storage devices: ACS Appl. Mater. Interfaces, 8, 15449 (2016); Scientific Reports, 7, 41981 (2017)., (d) energy generator devices: Adv. Mater., 27, 2866 (2015); Adv. Mater., 27, 4744 (2015); J. Mater. Chem. A, 1, 8541 (2013); Int. J. Hydro. Energy, 39, 7422 (2014). (e) sensors: Nano Letters, 15, 5240 (2015). (f) flexible display: Nanoscale, 9, 1978 (2017). (g) PM filter: Nano Letters, 17, 4339 (2017), (h) memory device: Adv. Funct. Mater., 27, 1701138 (2017).] needs to be discussed in the introduction part.

7) Ag NW usually has passivation coating such as PVP. In the junction between nanowires, the PVP layer should play an important role. Any discussion?

8) Why is there a hysteresis in IV curve, Figure 1(b)? and does it follow the exactly same curve for repeated test under same current condition (no aging effect)?

9) Some acronyms need full name, for example, CF RRAM.

Reviewer #3 (Remarks to the Author):

This manuscript claims the discovery of Winner-Take-All (WTA) paths in certain random nanowire network. But I do not think the present manuscript has provided sufficient evidence or reasoning for the claim.

(1) The authors correlate the WTA paths with the key characteristic of conductance plateaus in the log-log plot of conductance against current compliance. I do not know why. As the conductance of Ag nanowire junctions clearly increases with current compliance (Figure 1c), why that of WTA paths is independent of current compliance?

(2) The claim of observing conductance plateaus in Figure 1(d) is not persuasive because all the plateaus are actually restricted within very small regions. For me, the whole Figure 1(d) looks more like a nonlinear I-V curve with acceptable noise level that is a typical characteristic of percolating nanowire systems. Even if I agree that there are plateaus in Figure 1(d), why the second (upper) plateau has much higher conductance than the first (lowest) one? Does that imply that the second WTA path is much stronger than the first one? Then how could the first WTA path win the second one in the beginning?

(3) The authors claim (Lines 235-236) the conductance plateaus can be observed only when extremely fine current steps are used in the measurements. This is consistent with Figure 1(d), but not consistent with their claim in Figure 4 where the first WTA path prevails for current compliance of both 50 nA (Figure 4b) and 500 nA (Figure 4c). That means the plateaus should be observed over the current range of more than one order of magnitude.

(4) Based on my comment (3), I am not sure whether the observed percolating path in Figure 4 should be identified as WAT path. Any predominant percolating path should have the same behavior. Figure S8 seems not informative, and the scale bars of 2 μm make all the Ag nanowires look more like "micro-wires".

(5) The computation part well predicts the step-wise plateaus, but it seems not insightful as there is no discussion about the mechanism or driving force of the WTA paths.

Therefore, I cannot see any solid evidence for the formation of WAT paths. This work in the present version is of little interest to the society. Instead, the authors should attain multiple and consistent data before drawing a conclusion. The claims based on single data (e.g., Figure 1d and Figure 4) are not persuasive.

Reviewer #4 (Remarks to the Author):

The manuscript "Emergence of winner-takes-all Connectivity Paths in Random Nanowire Networks" authored by Manning et al. is an interesting and novel work discussing the conductance evolving behavior of network with variable current compliance and showing behavior of neuromorphic nature and quantum conductance. The work is expected to be interesting to broad readers working on electrical percolation, neuromorphic devices, neural network etc. Work involves experimental demonstration as well as numerical simulations. Thus, we strongly suggest for its acceptance. However, there is good room for its improvement for broader readership. Following are our questions and suggestions which should be added in manuscript for more clarity.

The measurement done in Figure 1 and S1 is not clearly explained. It should be made more detailed. The behavior of Figure 1(b) and S1(b) should also be explained.

Trend appears to be opposite for 200 pA to 800 pA in Figure 1(b). Authors may clarify.

Line 100: "However not all networks exhibit conductance plateaus; in some instances, networks fail electrically in the PL scaling regime. But some networks such as AgNWNs always exhibit plateaus."

The above line is not very clear, what so specific about AgNWNs and what problems other networks face.

The resistance includes wire resistance and junction resistance. While the junction resistance is updated on applying current, the inner resistance will remain the same. Will it not affect the overall resistance?

What are the experimental values of junction resistance and wire resistance for the case studies. The current dynamics are controlled by power (α) and prefactor (A). What experimental conditions govern these parameters.

In the numerical studies, authors can compare frequency distribution of conductance in different wires for different regimes. It can easily be done from their current map data and will reflect which regime takes all phenomena. Refer Figure 3 of J. Appl. Phys. 122.4 (2017) 045101.

There are three cases $\alpha_j < 1$, $\alpha_j = 1$ and $\alpha_j > 1$. Why all junctions of networks will lie in same regime.

The conductance jumps typically are sharply seen why in the present experimental work there are not so sharp. What can be the reason? Refer Figure 2 of Frank, Stefan et al. Science 280.5370 (1998): 1744-1746.

Can the Y axis of 1(d) be made in terms of G_0 , it will more clearly show the quantized jumps.

Authors may show these measurements at lower temperature to reduce the noise.

Authors may cite relevant papers for improving the discussion such as J. Appl. Phys. 122.4 (2017) 045101 and Physical Review B 86.13 (2012): 134202 which deal with current maps and current distribution in NW networks.

Line 226 : $\rho_p = \rho_0 / n$. What is typical number of n (no. of active junctions) in experimental system. Experimentally, plateau was observed at $\rho_0 / 8$. Is the system size has $n = 8$.

Please find attached our paper “Emergence of Winner-takes-all Connectivity Paths in Random Nanowire Networks” which we wish to resubmit to Nature Communication following receipt of the referee’s reports. At the outset, we would like to acknowledge the work of the Editorial team at Nature Communications and the referees for their valuable feedback. Their suggestions and comments have significantly improved our manuscript. Below, we address all comments raised by the referees on a point by point fashion and describe how each is dealt with in the revised manuscript (list of revisions at the end of our response).

Answers to Reviewer #2

We would like to thank the reviewer for the encouraging comments and valuable feedback. In particular, we greatly welcome the very first comment of the reviewer – who is an expert in the field:

“1) As a researcher in metal nanowire network, I have never thought about this kind of WTA conducting path formation. The authors very nicely provided extensive discussion and evidences to support their claims.”

In the following, we address each point raised by the reviewer:

“2) However, the authors said, “These observations are expected to have important implications for the operation of neuromorphic devices based on reservoir computing” without any brief explanation on the ‘reservoir computing’. Short explanation on ‘reservoir computing’ will greatly help the readers better understand the main idea in this paper.”

Reservoir computing is a neural network-based strategy for processing time-varying inputs. While the connectivity structure of the network or reservoir remains fixed, the nodes (the junctions in the case of NWNs) evolve dynamically in response to input signals and collectively define the internal state of the reservoir. This serves to map lower-dimensional input signals onto outputs of higher dimensions, which are then examined by an external readout function. Reservoir computing is highly effective for identification, prediction and classification tasks. We included this short explanation in the new version of the manuscript.

“3) Even very tiny amount of current can induce a dramatic temperature rise in a metal nanowire (Small, 10(24), 5015-5022 (2014)). The junction resistance between metal nanowires plays a very critical role. Due to very small size, tiny current flow during measurement step can induce localized resistive heating at the nanowire junction, which leads to the welding effect at junction. Once there is any welding phenomena happens the electrical properties of the metal nanowire junction will change irreversibly. Especially, for the WTA case, there will be current crowding at the junctions along the WTA path and resistive heating effect at the junction and along the nanowires as well (just like the pictures shown in Figure 3). Any discussion on this effect during measurement?”

Junction welding does not occur in our case since the current levels used to establish the WTA path are in the nA to μA range and distributed across networks that are 50 - 500 μm in size. It is true that the flow of current through nanoscale junctions can induce a large Joule heating effect which causes irreversible morphological changes to the junction such as the welding effect, which the reviewer mentioned. Increasing the current levels to the mA range would most certainly cause an irreversible change in the pathway of the current flow, this is highlighted in the revised manuscript. The formation of nano-welds in Ag NW junctions has been previously shown to dramatically reduce the contact resistance [V. V. Radmilovic, et al., Nanotechnology **28**, 385701 (2017)]. Indeed, our group has previously compared the effectiveness of electroforming and annealing single nanowire junctions with SEM images of electroformed junctions (to a compliance limit of 10 μA) showing no signs of nanowire melting/fusing [A. T. Bellew, et al., ACS Nano **9**, 11422 (2015)]. In the Yeo et al. reference mentioned by the reviewer, current levels in the mA range were used to drive the nano-heater. In another work by Song et al. ACS Nano **8**, 2804 (2014), Joule heating effects and electromigration of a single nanowire junction was observed with current densities of $\sim 1.5 \times 10^7 \text{ A/cm}^2$ ($\sim 150 \mu\text{A}$ drive current) and when almost 200 mA was passed through a 1.5 x 1.2 cm^2 network.

“4) Does this WTA conducting path in random nanowires happen to the nanowires network with good welding (thermal annealing)? or only to the NWN without welding?”

The formation of the WTA conducting pathway in random Ag NWN only occurs when the network self-selects the conducting pathways through junctions which are initially insulating. NWNs subjected to thermal annealing are highly conductive and therefore typically contain multiple low-resistance pathways between the electrodes, as evidenced by Sannicolo et al. Nano Letters **16**, 7046 (2016) where the onset of electrical conduction was visualized by Infra-Red Thermography.

“5) Some studies presented that NW length is very important for the electrical properties of metal nanowire network (Nanoscale, 4, 6408 (2012); Crystal Growth & Design, 12, 5598 (2012)). This WTA phenomena could be dependent on the NW geometry such as NW diameter and NW length and deposited NW density. Any discussion on this?”

The referee is correct that the WTA depends on the NW geometry. Increasing the wire length leads to two different contributions that can, in principle, impact the appearance of WTA paths. First, the network connectivity is sensitive to changes in the wire length, since the number of junctions per wire is proportional to $L^2\rho_w$ [see O’Callaghan et al. PCCP **18**, 27564 (2016)], with L being the wire length and ρ_w being the wire density. This connectivity determines the wire segment lengths used in the definition of the intra-wire resistance $R_{in} = \rho l/A_c$ where ρ is the wire resistivity, l its segment length, and A_c its cross-sectional area. However, as mentioned in section 6 of the supplemental information, the WTA arises in the limit of $R_{in} \ll R_j$ and in this limit the influence of R_{in} on the nanowire geometry will have only a minor impact on the WTA effect. Another contribution comes from the number of wires (n_{wta}) contained in the WTA path spanning the electrodes. This number is proportional to the ratio W/L , where W is the electrode separation and L the wire length, and it plays a direct role in determining the conductance value of the WTA plateau at Γ_0/n_{wta} . A brief explanation on this point is added in the new version of the manuscript.

“6) Metal nanowire percolation networks have shown various applications. The detailed example of metal nanowire will be very helpful for readers. The application areas on metal nanowire percolation network [(a) touch panel: Adv. Funct. Mater., 23, 4171 (2013); RSC Advances, 6, 57434 (2016); Nanotechnology, 27, 295201 (2016). (b) stretchable electrode: Adv. Mater., 26, 5808 (2014); Adv. Mater., 24, 3326 (2012); Adv. Mater., 27, 6397 (2015); , (c) energy storage devices: ACS Appl. Mater. Interfaces, 8, 15449 (2016); Scientific Reports, 7, 41981 (2017)., (d) energy generator devices: Adv. Mater., 27, 2866 (2015); Adv. Mater., 27, 4744 (2015); J. Mater. Chem. A, 1, 8541 (2013); Int. J. Hydro. Energy, 39, 7422 (2014). (e) sensors: Nano Letters, 15, 5240 (2015). (f) flexible display: Nanoscale, 9, 1978 (2017). (g) PM filter: Nano Letters, 17, 4339 (2017), (h) memory device: Adv. Funct. Mater., 27, 1701138 (2017).] needs to be discussed in the introduction part.”

We acknowledge the relevance of the applications pointed out by the referee and have modified our introduction to account for this wide range of applications and have incorporated the associated references.

“7) Ag NW usually has passivation coating such as PVP. In the junction between nanowires, the PVP layer should play an important role. Any discussion?”

The PVP layer is extremely significant for the emergent phenomena observed in our work. As discussed in point 5, the contrast between the metallicity of the core wires and the pristine insulating nature of the interwire junctions - due to the PVP shell - plays a crucial role in the formation of conductive paths in nanowire networks. The PVP shell works as a “current regulator” or filter that does not let the whole network activate abruptly as it is electrically stressed. The PVP shell enables the adiabatic growth of nanoscale conductive filaments across it via a soft breakdown mechanism. In this way, junction resistances can be gradually reduced as the current compliance is increased. In samples where the PVP layer was very thin or removed entirely, the network failed to activate gradually and WTA paths were not observed.

“8) Why is there a hysteresis in IV curve, Figure 1(b)? and does it follow the exactly same curve for repeated test under same current condition (no aging effect)?”

The hysteresis is due to the nonlinear increase of the current across the network as charge is leaked through the highly resistive junction of two overlapping PVP coated Ag nanowires which form a Metal-Insulator-Metal junction. The current levels for these sweeps are quite low (pA to nA); in this current range, the hysteresis loops are repeatable, although the size of the loops become smaller as the junction conductance increases with current flow. This is illustrated in the figure below, for a single Ag NW junction, as the current compliance level is increased, and sweeps are run in quick succession, similar shaped I-V curves are obtained. At higher current compliance, the hysteresis loops collapse. The same behaviour is observed for Ag NWNs. However, if the system is allowed to relax for a period of time the hysteresis loop will open again due to the dissolution of the CFs - the decay and retention of the conductance levels in these systems is currently under investigation and will be reported in a future manuscript. These plots and a discussion of the repeatability of the I-V curves has been added to the supplemental information.

“9) Some acronyms need full name, for example, CF RRAM.”

CF RRAM stands for conductive filament-based resistive random-access memory. We have provided full names for all acronyms in the new version of our manuscript.

Answers to Reviewer #3

The reviewer states:

“This manuscript claims the discovery of Winner-Take-All (WTA) paths in certain random nanowire network. But I do not think the present manuscript has provided sufficient evidence or reasoning for the claim.”

We hope that our point-by-point explanations and clarification in addition to the significant revisions made in our manuscript will allay the reviewer’s concerns and address all criticisms. We also would like to direct the reviewer to the revised supplemental material which contains a substantial amount of information and additional supporting data for all our findings.

“(1) The authors correlate the WTA paths with the key characteristic of conductance plateaus in the log-log plot of conductance against current compliance. I do not know why. As the conductance of Ag nanowire junctions clearly increases with current compliance (Figure 1c), why that of WTA paths is independent of current compliance?”

The signature of the formation of a WTA path is the presence of a subquantum conductance plateau that represents the point where there is a change in the conduction mechanism of the junctions that make up the WTA path. The actual value of the first conductance plateau depends on the number of junctions that comprise the WTA path. Up to that point the junction conductances increase with current compliance (as the reviewer notes), but afterwards these junctions become ohmic and exhibit a conductance that is independent of current, i.e., a plateau, which is in fact a signature of ohmic behaviour. A conductance plateau means that $\alpha_j = 0$, so that the power dissipation given by equation (7) in the main text scales as the current squared, consistent with ohmic conduction. At current compliances beyond the first plateau other parts of the network begin to contribute to the current transmission. This explanation is emphasised in the revised manuscript.

“(2) The claim of observing conductance plateaus in Figure 1(d) is not persuasive because all the plateaus are actually restricted within very small regions. For me, the whole Figure 1(d) looks more like a nonlinear I-V curve with acceptable noise level that is a typical characteristic of percolating nanowire systems. Even if I agree that there are plateaus in Figure 1(d), why the second (upper) plateau has much higher conductance than the first (lowest) one? Does that imply that the second WTA path is much stronger than the first one? Then how could the first WTA path win the second one in the beginning?”

In Figure 1(d), which is not an I-V curve but a curve displaying the systems conductance over a range of current compliance limits, the first plateau occurs at $\Gamma_p \sim 1.0 \times 10^{-5}$ S whereas the second and third plateaus occur at $\Gamma_2 \sim 1.3 \times 10^{-5}$ S and $\Gamma_3 \sim 1.6 \times 10^{-5}$ S, respectively. The WTA represents the most primitive conductive path formed in the network and it exhibits the lowest conductance plateau value. The ratio Γ_p/Γ_0 gives 1/8 which means that the particular network investigated in Figure 1(d) has the equivalent conductance of 8 quantum conductors in series. As the network continues to be (electrically) stressed, other conductive paths will be activated and so the network can exhibit additional conductance plateaus. Assuming the paths are independent and operate in parallel, the measured conductance is the sum of the individual path conductances. In this regard, the conductance of the path associated with the first plateau is clearly the highest, as shown in Figure 1(d), which has been revised to allow the reader to better visualise the conductance plateau values relative to Γ_0 . Our simulations (cf. Figure 3) show, however, that new emergent paths can co-opt junctions that were optimised in earlier paths. Moreover, part of the overall increase in conductance beyond the first plateau is due to a slow but gradual increase of the conductance of all the junctions across the network as shown in Figure 3.

“(3) The authors claim (Lines 235-236) the conductance plateaus can be observed only when extremely fine current steps are used in the measurements. This is consistent with Figure 1(d), but not consistent with their claim in Figure 4 where the first WTA path prevails for current compliance of both 50 nA (Figure 4b) and 500 nA (Figure 4c). That means the plateaus should be observed over the current range of more than one order of magnitude.”

The WTA emerges at the first conductance plateau but remains even after the emergence of additional plateaus and paths (the gradual electrical stress does not destroy the initial path). This is the basis for the analysis of the conductance plateau in our response to question 2 above, and is consistent with our simulations (Figure 3) and experiments (Figure 4). This does NOT mean, however, that the current plateau should exist from 50 nA to 500 nA. The additional paths that are emerging at the bottom electrode to the left of the WTA path contribute to an increase in conductance, as do the paths emerging from the sides of the upper electrodes. We emphasise this important point in the revised manuscript.

“(4) Based on my comment (3), I am not sure whether the observed percolating path in Figure 4 should be identified as WAT path. Any predominant percolating path should have the same behaviour. Figure S8 seems not informative, and the scale bars of 2 mm make all the Ag nanowires look more like “micro-wires”.”

We appreciate that the path contrast shown in Figure 4(b) does not correspond exactly to the simulated case shown in Figure 3, and may be the source of some confusion. To explain this discrepancy, it is worth highlighting the difference between the predicted and observed WTA paths. In experiments, wires located closer to the electrodes are optimized faster than those in the middle of the network, allowing these quickly-activated wires to effectively become an extension of the electrode. The reason for this is that the junctions within the network are essentially twice the thickness of the PVP layer, whereas for wires that directly touch the electrode, the junction is just the thickness of a single PVP layer. Because of this, a WTA path tends to have an effective length that is reduced compared to the full separation between the electrodes (note in Figure 4(d) a WTA path with no branches being formed at the half-bottom part of the panel). The computational model used throughout the manuscript does not differentiate between junctions near the electrodes and elsewhere. Our new Figure S11 (also depicted below) shows, that when we account for the facile activation of junctions in the vicinity of the electrodes in our simulations, we find an increase in the numbers of activated wires at the electrodes in contrast to the clean path seen in Figure 3. This however, does not in any way change the conclusions of the paper and the conditions required for WTA formation

We also corrected the scale bar information appearing in the caption of Figure S11 (now Figure S14) in the supplemental material.

“(5) The computation part well predicts the step-wise plateaus, but it seems not insightful as there is no discussion about the mechanism or driving force of the WTA paths.”

Our experimental finding of self-similar scaling and the emergence of conductance plateaus is undoubtedly surprising given the disordered nature of these networks. However, from this observation alone it would not have been possible to infer the way current flows through these networks. This was only possible through simulations. In fact, the insights provided by the simulations were essential: (i) demonstrate that plateaus only arise when $\alpha > 1$; (ii) demonstrated that the plateaus in Figure 3 correspond to WTA paths; (iii) establish that the plateaus correspond to a fraction of the quantum of conductance. Finally, these computational findings played a key role in the experimental search for the WTA paths seen in Figure 4 of the manuscript.

In a future manuscript we will consider the atomistic mechanism that underlies the power law scaling demonstrated here. Such a study is clearly outside of the scope of the present manuscript.

“(6) Therefore, I cannot see any solid evidence for the formation of WAT paths. This work in the present version is of little interest to the society. Instead, the authors should attain multiple and consistent data before drawing a conclusion. The claims based on single data (e.g., Figure 1d and Figure 4) are not persuasive.”

This work provides for the first time an understanding of how the properties (in this case the conductivity) of a network material can be controlled by regulating the connectivity within, which is important for applications such as transparent conductors and hardware implementations of neural networks. The fact that the junctions don't immediately activate, but form a WTA path makes these networks capable of embedding information that can be recalled for use in memory, logic and possibly analogue computation.

Our results are not based on single data. Restrictions in word count limit the main text to the key results that corroborate our claims on the formation of WTA paths. We extended the already substantial material presented in the supplemental information [see Figures S2, S4, and S11] to clearly demonstrate that our findings are based on numerous experiments conducted on an ensemble of samples and I-V characterizations.

Answers to Reviewer #4

We would like to thank the reviewer for his/her positive comments and strong recommendation for publication in Nature Communications:

“The manuscript “Emergence of winner-takes-all Connectivity Paths in Random Nanowire Networks authored by Manning et al. is an interesting and novel work discussing the conductance evolving behavior of network with variable current appliance and showing behavior of neuromorphic nature and quantum conductance. The work is expected to be interesting to broad readers working on electrical percolation, neuromorphic devices, neural network etc. Work involves experimental demonstration as well as numerical simulations. Thus, we strongly suggest for its acceptance. However, there is good room for its improvement for broader readership. Following are our questions and suggestions which should be added in manuscript for more clarity.”

In the following, we address each point raised by the reviewer:

“(1) The measurement done in Figure 1 and S1 is not clearly explained. It should be made more detailed. The behavior of Figure 1(b) and S1(b) should also be explained. Trend appears to be opposite for 200 pA to 800 pA in Figure 1b). Authors may clarify.”

In response to the reviewers concerns we have revised Figures 1 and introduced new Figures S1 and S2, we have also described the nature of the hysteresis in Figure 1(b) and detailed the electrical measurement of Figure S1(b). In Figure 1(d) the 800pA (red trace) does indeed reach the compliance current at a slightly higher voltage than the 200pA black trace, this is due to a slight variation in activation voltages which is most pronounced at low current compliances (pA to nA range). To clarify this figure, we have replaced the 200 pA sweep with the first sweep run on this nanowire junction at a compliance current of 100 pA. A further explanation and discussion on the origin of the hysteresis loops can be found in our response to reviewer 2 (Q8) and included as a revised Figure S2.

“(2) Line 100: “However not all networks exhibit conductance plateaus; in some instances, networks fail electrically in the PL scaling regime. But some networks such as AgNWNs always exhibit plateaus. “ The above line is not very clear, what so specific about AgNWNs and what problems other networks face.”

We have studied a range of systems: Ag NWNs made of core-shell Ag-PVP wires, Ni NWNs made of core-shell Ni-NiO wires, and Cu NWNs made of core-shell Cu-CuO. The PVP coating allows charge to leak through the junction by a number of different mechanisms, depending on the magnitude of the current flow; it begins as tunnelling followed by electromigration, and finally current induced joule heating, which can ultimately weld the junctions together if subjected to extremely high current densities. This allows the slow ripening of each junction in the pathway, which we show in the manuscript to proceed for $\alpha_j > 1$. For this reason, the conductance plateaus are easier to observe in the Ag-PVP NWNs. Cu is another highly electroactive material and its exponents were found to be in the critical regime fluctuating around 1 (see updated table S1 in the supplemental information). We have observed conductance plateaus for Cu NWNs when the current compliance is increased in very precise increments, this has been added to the supplemental information [see new Figure S4 also shown below]. In the case of Ni, α_j was also found to fluctuate around 1 but no plateaus were observed.

“(3) The resistance includes wire resistance and junction resistance. While the junction resistance is updated on applying current, the inner resistance will remain the same. Will it not affect the overall resistance? What are the experimental values of junction resistance and wire resistance for the case studies.”

The power law scaling behaviour is only found for junction dominated systems in which the inner resistances R_{in} are much smaller than R_j , where R_j is the junction resistance. This condition is met for most of our NW systems. For example, in Ag-PVP NWNs, the average inner resistance of the wires is about 5Ω whereas junction resistances are of the order of $k\Omega$ s. In Ni-NiO NWNs, the average inner resistance of the wires can be three times larger than for Ag-PVP NWNs. In the same way, junction resistances in Ni-based NWNs can also be three to four times larger than for Ag-PVP NWNs. The junction resistance. The experimental values for R_j and R_{in} were taken from our previous publication Ref. 33 ACS Nano **9**, 11422-11429 (2015).

“(4) The current dynamics are controlled by power (α) and prefactor (A). What experimental conditions govern these parameters.”

The intrinsic microscopic factors associated with the exponent α and the prefactor A in nanojunctions are briefly discussed in section 2 of the supplemental material in which a physical interpretation for these parameters is extracted by analogy with the ion-drift model. There we demonstrate that the prefactor A depends on the mobility of the charge carriers, the decay rate of the tunnelling barrier, and the width of the junction [cf. equation (s8)]. The exponent accounts for the nonlinear effects taking place in the conducting channel subjected to extremely intense electric fields ($\sim 10^7$ V/cm). Nonetheless, we recognize that more work is necessary to establish a more accurate relation between our phenomenological power law description and the microscopic mechanisms involved during filament growth. For now, our scaling picture follows a phenomenological bottom-up approach in which we make use of the available experimental data taken from nanojunction measurements to make predictions on the overall behaviour of the macro-network. For this reason, we are investing on the development of a microscopic description based on Kinetic Monte Carlo method with the aid of shedding light on the interpretation of the power law parameters.

“(5) In the numerical studies, authors can compare frequency distribution of conductance in different wires for different regimes. It can easily be done from their current map data and will reflect winner takes all phenomena. Refer Figure 3 of J. Appl. Phys. 122.4 (2017) 045101.”

We acknowledge the referee’s suggestion; we would like to point out that this information is self-contained in the current-heat snapshots and animations provided. In this work, we opted to present the calculated data in a format that reveals the formation of the WTA pathways explicitly. The current map data can be easily converted into frequency distributions from which the WTA phenomenon can be also represented. We already have investigated such distributions and observed that the pathway transport phenomenon in random NWNs relates to the spread of the conductance/current distribution, i.e. networks at the WTA state yield sufficiently narrow distributions. As the networks enter multi-path conduction in the post-power-law regime, the shape of the distributions is highly affected as it tends to spread.

The reference mentioned above by the reviewer was included in the new version of the manuscript.

“(6) There are three cases $a_j < 1$, $a_j = 1$ and $a_j > 1$. Why all junctions of networks will lie in same regime?”

The junctions in a given network won’t all fall into the same category of exponents. This is just an idealized case we discuss in the manuscript for the sake of simplification. Nonetheless, we find that different surface layers exhibit distinct values of α_j and that there is a distribution of α_j values for a given wire system (see Table S1 in the supplemental material). We also simulated the effects of a dispersion in the exponents and these are shown in Figure 2(d) and section 5 of the supplemental material. In fact, any sort of dispersion (in the exponents and prefactors) can play an important role in the conductance scaling of the networks and an analysis of this behaviour is also in progress.

“(7) The conductance jumps typically are sharply seen why in the present experimental work there are not so sharp. What can be the reason Refer Figure 2 of Frank, Stefan et al. Science 280.5370 (1998): 1744-1746.”

The sharp plateaus predicted by the simulations and which are evident in the reference supplied are a result of the hard cut-offs we imposed for the junctions allowing them to optimize only within the range of $[R_{\text{off}}, R_{\text{on}}]$ with $R_{\text{off}} = 10^4 \text{ k}\Omega$ and $R_{\text{on}} = 12.9 \text{ k}\Omega$. In the network experiments, a number of junctions can optimize further than $12.9 \text{ k}\Omega$ through the formation of additional conductance channels and this would make the plateaus less sharp. Note that it is extremely difficult to access the maximum conduction capacity of a junction given the complexity of the dynamical/stochastic mechanisms occurring during filament formation and the highly-disordered environment these units experience when assembled on a random network frame. Other factors such as dispersion in the exponents can lead to more than one path being optimized simultaneously (with one path slightly ahead from the others) and this also softens the shape of the plateaus. Optimization of interconnected paths such as that depicted in Figure S8 in the supplemental material also affects the sharpness of the plateaus.

“(8) Can the Y axis of $I(d)$ be made in terms of G_0 , it will more clearly show the quantized jumps.”

We have modified the Figure 1(d) in terms of Γ_0 so that the values are more clearly accessible.

“(9) Authors may show these measurements at lower temperature to reduce the noise.”

From our preliminary results considering temperature variations, we observe that Ag NWNs show a transition from a metallic to a more semiconducting behaviour at low temperature, similar to the activation dependent transition observed by Oliver et al. Appl. Phys. Lett. **109**, 203101 (2016) for Ni NWNs. These are fascinating results which may provide additional insight into the mechanisms by which the junctions activate but are beyond the scope of this manuscript where we focus our attention on the scaling process and the formation of WTA paths.

“(10) Authors may cite relevant papers for improving the discussion such as J. Appl. Phys. 122.4 (2017) 045101 and Physical Review B 86.13 (2012): 134202 which deal with current maps and current distribution in NW networks.”

We acknowledge the referee’s suggestions and included both mentioned references in the new version of our manuscript.

“(11) Line 226 : $\Gamma_p = \Gamma_o / n$. What is typical number of n (no. of active junctions) in experimental system? Experimentally, plateau was observed at $\Gamma_o / 8$. Is the system size has $n = 8$?”

Yes, $\Gamma_o/8$ indicates that the WTA path of the network contains 8 optimized junctions in series for the system studied in Fig 1(d). The number n can be estimated as a first approximation from the source-drain electrode separation D and the average length of the wires L . An idealized case in which $n-1$ wires of length L joined end-to-end in series over the inter-electrode separation D , we would obtain $n=1+D/L$ (this includes the two connections the 1st and the $(n-1)$ th wire make with the electrodes). However, this is an extremely rough estimation which assumes a most unlikely linear arrangement of wires in the WTA state. We should, however, emphasise in experiments (especially those conducted in samples of large dimensions, typically above 100 x 100 μm), wires located closer to the source electrode tend to activate much faster than in the middle of the network making these quick-activated wires a sort of extension of the source electrode (see our response to Reviewer 3, question 4). In this way, a real WTA path tends to have an effective length that is reduced in comparison to the full source-drain electrode separation (see Section 5 of supplementary materials) and the value of n is determined by this effective length.

LIST OF CHANGES:

1. Brief explanation on reservoir computing was included in the last paragraph of the introduction (page 3) as requested by reviewer #2:

“These observations are expected to have important implications for example in the area of neuromorphic devices based on reservoir computing [29, 30, 31], a neural network-based strategy for processing time-varying inputs which is highly effective for identification, prediction and classification tasks [32].

[32] While the connectivity structure of the network or reservoir remains fixed, the nodes (the junctions in the case of NWNs) evolve dynamically in response to input signals and collectively define the internal state of the reservoir. This serves to map lower-dimensional input signals onto outputs of higher dimensions, which are then examined by an external readout function.”

2. Modification in the introduction (page 2) highlighting the broad spectrum of applications nanowire network materials can take part of including all relevant citations.

“Thermal, mechanical and chemical processes have been utilized to minimize these barriers and develop various applications based on metal nanowire networks (NWN) [1, 2] including flexible and transparent conductors [3, 4, 5, 6, 7, 8, 9, 10], energy storage [11, 12] and generator devices [13, 14, 15], sensors and memory devices [16, 17].”

3. Numerous references added in the main manuscript, those include [3] to [9], [11] to [17], and [43], [44], and [45].

4. A sentence about the welding effect in the nanowire connections was included in the first paragraph of the RESULT section (page 5) as mentioned by reviewer #2:

“It is also important to note that all NWN samples studied in this work are 50 to 500 μm in size and experience current levels in the nA to μA range so as to avoid junction welding due to Joule heating.”

5. Discussion on how the nanowire geometrical features impact the PL dynamics and formation of WTA paths was added on page 14 as requested by reviewer #2:

“To demonstrate this effect, we artificially increased the resistivity of the wires while keeping the junction resistances fluctuating in the same $[R_{off}, R_{on}]$ range. These results are shown in Figure S12 in the supplemental material and they indicate that a clean PL scaling behaviour and a conducting WTA state can only be found for systems in which $R_{in} \ll R_j$. Other sample characteristics such as NW geometrical aspects, network density, width of the devices, contact resistance (cf. Figure S11 in the supplemental material), and electrode separation are also found to affect the PL dynamics and formation of WTA paths in random NWNs because they all have an impact on the network connectivity [43, 44, 45].”

6. New Figure S2 added in the supplemental material to clarify the I-V characterization made in our NW systems and demonstrate how the experimental conductance versus compliance current curves were obtained.
7. Acronym “CF RRAM” written down explicitly on page 6 as pointed out by reviewer #2.
8. Figure 1(d) as well as its respective caption were modified as requested by reviewer #4 and to better answer reviewer #3 question about the order of the conductance plateaus.
9. Figure 4 now contains the current levels written on the panels and its caption includes the measured conductance values for each one of the PVC images.
10. New Figure S11 in the supplemental material depicting the binary activation of a NWN following the assumption that nanowires in contact with the source electrode will activate faster than the rest of the wires. This result was included to demonstrate reviewer #3 that our models can incorporate any level of complexity to account for more realistic conditions during the formation of WTA paths.
11. Caption of former Figure S11 (now Figure S14) in the supplemental information was corrected as pointed by reviewer #3. Current levels were included on the panels and the resolution of the figure was improved.
12. Values of exponents and prefactors presented on table S1 in the supplemental material were updated.

REVIEWERS' COMMENTS:

Reviewer #2 (Remarks to the Author):

The authors responded well to the reviewers' comments. This should be ready for publication.

Reviewer #3 (Remarks to the Author):

The revision has not set aside my major concern about the relation between the WTA paths and the observed conductance plateaus. In my opinion, there is no convincing evidence for the existence of WTA paths and this manuscript is not ready for publication.

(1) It looks that two prerequisites are needed for the conductance plateaus in the nanowire networks. One is the junctions between any two nanowires are saturated at certain conductance (or quantum conductance as claimed in the manuscript). The other is the WTA keep the number of junctions in the percolating path(s) constant. The former is modeled by the authors with the PL+C model in Fig. S5. However, the experimental results do not support such a behavior (modeling). Looking at Fig. 1(c), I cannot see any cut-off conductance for the Ag nanowire junctions (Ag NW Jxn). Instead, their conductance continuously increases with the current compliance in power laws even when the value is far above the quantum conductance. So, as the "individual" Ag NW junctions do not have any cut-off conductance, how should the "networks" contain the conductance plateaus? This is actually the question in my original Comment (1).

(2) It is still very difficult for me to accept that there are "intrinsic" (or fixed) conductance plateaus for the nanowire networks in Figs. 1(c), 1(d) and S4. As the so-called plateaus are only restricted within very narrow ranges of current compliance (except that in Fig. S4(a)), any measurement noise or sample variation could also lead to such behaviors. Therefore, systematic statistics over different measurements and samples should be essential for this research. Although in their response to my previous comments, the authors claim their results are not based on single data, but on numerous experiments, I did not see any statistical analysis in the manuscript.

(3) The authors claim (Lines 242-243) that the conductance plateaus can be observed only when "extremely fine" current steps are used in the measurements. This is consistent with their simulation results in Figs. 2, 3 and S12. However, the experimental results in Fig. S4(a) indicate a plateau over more than one order of magnitude of the current compliance. Could this imply the observed conductance plateaus may result from different mechanism from the WTA paths?

(4) In Fig. 1(d), the authors indicate three plateaus with conductance equal to fractions (1/8, 1/6 and 1/5) of quantum conductance. Are the fractions just accidental or intrinsically resulting from some underlying physics? Above the three lowest plateaus, there look several more plateaus in Fig. 1(d). How much is their conductance? Are they also fractions of quantum conductance?

Reviewer #4 (Remarks to the Author):

I find that the authors have well addressed the queries raised by the referees.
I have no further comments. The manuscript is now acceptable.

Please find attached our paper “Emergence of Winner-takes-all Connectivity Paths in Random Nanowire Networks” which we wish to resubmit to Nature Communication following receipt of the referee’s reports and the editorial format requirements. We would like to acknowledge the work of the Editorial team at Nature Communications and the referees for their valuable feedback. Below, we address all comments raised by referee #3 on a point by point fashion and describe how each is dealt with in the revised manuscript (list of revisions at the end of our response). In the following, we present an editing checklist that we conducted in the manuscript source files in order to comply with the editorial format requirements of Nature Communications.

Answers to Reviewers #2 and #4

We would like to thank the reviewers for the encouraging comments and for acknowledging our efforts in delivering an improved version of our work.

Answers to Reviewer #3

The reviewer states:

“The revision has not set aside my major concern about the relation between the WTA paths and the observed conductance plateaus. In my opinion, there is no convincing evidence for the existence of WTA paths and this manuscript is not ready for publication.”

It is unfortunate that our message is yet not clear for the reviewer. Our main evidence of WTA paths comes in a very explicit form, by means of the PVC images of networks interrogated by electrical signals in which only current-carrying wires appear in dark contrast. Independent on which conductance level the network might be located in those images, they reveal that only a few wires participate in the conduction process rather than the whole network and this is what we called WTA phenomenon in random nanowire networks. Certainly, the fact that these are highly disordered systems with transport properties ruled by undeterministic stochastic/microscopic mechanisms brings several complications to the problem. As a result, this work is a first attempt in providing a simplified (but yet accurate) picture of such complicated phenomena. We acknowledge that further investigation is required to refine our understanding on the complexity of these systems but at this stage, we felt that this first set of results is already substantial to be presented to the scientific community. In addition, we recognize that the term “WTA”, used in many other scientific fields, e.g. computer science, neuroscience, and data science, can reach out to distinct meanings and definitions. In this work, the term is used to refer to the conductive state registered in the PVC images showing the network using the least-resistance path to transmit sufficiently low current levels throughout its skeleton.

“(1) It looks that two prerequisites are needed for the conductance plateaus in the nanowire networks. One is the junctions between any two nanowires are saturated at certain conductance (or quantum conductance as claimed in the manuscript). The other is the WTA keep the number of junctions in the percolating path(s) constant. The former is modeled by the authors with the PL+C model in Fig. S5. However, the experimental results do not support such a behavior (modeling). Looking at Fig. 1(c), I cannot see any cut-off conductance for the Ag nanowire junctions (Ag NW Jxn). Instead, their conductance continuously increases with the current compliance in power laws even when the value is far above the quantum conductance. So, as the “individual” Ag NW junctions do not have any cut-off conductance, how should the “networks” contain the conductance plateaus? This is actually the question in my original Comment (1).”

The reviewer is right, the measurement in Fig. 1(c) does not manifest any conductance plateau at junction level and this is stated in the end of page 5 of our manuscript:

“Conductance plateaus were not observed in the case of single junctions as it is extremely difficult to control the conductance and stabilize the formation of single channel conductive filaments within a single junction.”

The PL+C junction model makes use of cutoffs for the junction resistances because this scheme is implemented on a network environment in which plateaus are observed. The incorporation of cutoffs also means that the resistance of a junction cannot be improved indefinitely; the junction can of course fail [as observed in the majority of single-junction measurements such as in Fig. 1(c)] or, in a network environment, it can stabilize in an optimum resistance state that we considered to be the quantum of conductance for the sake of simplification. Nonetheless, our model is flexible enough to assume any optimum resistance value for the junctions or even ensembles of values. Further analysis on this front is currently in progress.

“(2) It is still very difficult for me to accept that there are “intrinsic” (or fixed) conductance plateaus for the nanowire networks in Figs. 1(c), 1(d) and S4. As the so-called plateaus are only restricted within very narrow ranges of current compliance (except that in Fig. S4(a)), any measurement noise or sample variation could also lead to such behaviors. Therefore, systematic statistics over different measurements and samples should be essential for this research. Although in their response to my previous comments, the authors claim their results are not based on single data, but on numerous experiments, I did not see any statistical analysis in the manuscript.”

As we mentioned in our previous answer to the reviewers, this work presents a collection of high precision measurements conducted on a multitude of samples consisting of single -nanowires, -junctions, and networks. These measurements are made with high-resolution instrumentation capable of parsing ultra-low current levels with fA precision. However, it is a widespread problem within the nanoscience realm that many experiments involving the manipulation of nanoscale components (being those nanoparticles, nanowires, nanotubes, etc.) can be restricted to a relatively sparse ensemble of samples due to the ultimate challenges of working at such scale sizes can bring. Yet, we cared for synthesizing the largest number of samples possible and repeated the measurements over and over to reassure our conclusions.

“(3) The authors claim (Lines 242-243) that the conductance plateaus can be observed only when “extremely fine” current steps are used in the measurements. This is consistent with their simulation results in Figs. 2, 3 and S12. However, the experimental results in Fig. S4(a) indicate a plateau over more than one order of magnitude of the current compliance. Could this imply the observed conductance plateaus may result from different mechanism from the WTA paths?”

We discuss that plateaus occurring at sub-quantum of conductance range can indicate the emergence of a single WTA path given that the amount of junctions taking part of this path can be estimated from the fraction number of the plateau itself. However, we also argued in the manuscript that not every network material is prone for WTA conduction and the reviewer is right in this matter. In such cases, other conduction elements can indeed take part. For example, although our PL+C scheme assumes the quantum of conductance (G_0) as the ultimate state of a junction in a network, a junction can keep evolving and manifest multi-conductance channels, e.g. $2G_0, 3G_0, \dots$. Junctions can also fail during the network interrogation and so it can delay the evolution of an “in-progress” WTA path. In some cases, the dynamics of a given network with junctions of certain properties (e.g. with exponents ≤ 1) can lead to multiple paths evolving simultaneously rather than a dominant-evolving WTA path. We do not disregard any of these mechanisms that can undermine WTA conduction and we are not claiming that WTA is ubiquitous for every nanowire network material we worked so far. On the contrary, this work serves to distinguish the main physical features that can lead (or not) to the formation of WTA paths.

“(4) In Fig. 1(d), the authors indicate three plateaus with conductance equal to fractions (1/8, 1/6 and 1/5) of quantum conductance. Are the fractions just accidental or intrinsically resulting from some underlying physics? Above the three lowest plateaus, there look several more plateaus in Fig. 1(d). How much is their conductance? Are they also fractions of quantum conductance?”

The increasing conductance fractions highlighted in Fig. 1(d) corresponds to what we called in the text as post-power law (PPL) regime and its dynamics is found to be very susceptible to the characteristics of the junctions, in particular, on the exponent α_j . In the most idealized scenario pictured by the

simulations, the overall conductance of networks made with supra-linear junctions varies stepwise with the current in the PPL stage. Each conductance-jump corresponds to the opening of a new conducting path percolating the network as the sourced current increases. These paths can develop independently or they can superpose as depicted in Figures S8 and S9 in the supplemental material, respectively. Perfectly flat plateaus are not expected to be seen in the experimental results but as Fig. 1(d) evidences, discontinuities in the network conductance evolution can be noticed. The most meaningful fraction in the measurement is the one corresponding to the very first plateau, $\Gamma_p/\Gamma_0 = 1/8$, which evidences an effective network using 8 resistors in series, each one with a resistance of $1/\Gamma_0$, to propagate current. The second fraction at $1/6$ reveals that the network optimized its junctions in such a way that now it conducts with the efficacy of 6 resistors in series rather than 8. This does not mean that the network eliminated two resistors from its original WTA path; it means that the network improved its conduction profile by optimizing new junctions and/or opening up new conductive branches that now emulate the efficacy of 6 “quantum resistors” in series. In this way, these conductance fractions, equivalent to a simple circuit of resistors in series, serve as references to how the network enhances its transmission efficacy as current increases.